# Green Synthesis of Lutein-Based Carbon Dots Applied for Free-Radical Scavenging within Cells

**DOI:** 10.3390/ma13184146

**Published:** 2020-09-17

**Authors:** Dian Yang, Li Li, Lei Cao, Zhimin Chang, Qian Mei, Ruhong Yan, Mingfeng Ge, Chenyu Jiang, Wen-Fei Dong

**Affiliations:** 1School of Biomedical Engineering (Suzhou), Division of Life Sciences and Medicine, University of Science and Technology of China, Hefei 230026, China; yangdian@mail.ustc.edu.cn (D.Y.); cl1221@mail.ustc.edu.cn (L.C.); 2CAS Key Laboratory of Biomedical Diagnostics, Suzhou Institute of Biomedical Engineering and Technology, Chinese Academy of Science (CAS), Suzhou 215163, China; lil@sibet.ac.cn (L.L.); zhimin_chang@163.com (Z.C.); qmei@sibet.ac.cn (Q.M.); yrhzl@hotmail.com (R.Y.)

**Keywords:** carbon dots, fluorescence, ROS, radical scavenging activity, low cytotoxicity

## Abstract

Reactive oxygen species (ROS) in the body play an important role in various processes. It is well known that harmful high levels of ROS can cause many problems in living organisms in a variety of ways. One effective way to remove intracellular ROS is to use reducing materials that can enter the cell. Herein, we developed a strong reducing carbon nano-dot from a natural product, lutein, as an initial raw material. This is a hydrothermal synthesis method with the advantages of simplicity, high yield, mild reaction conditions, and environmental friendliness. The prepared carbon dots exhibit bright blue fluorescence, and have good water solubility and biocompatibility. In particular, the carbon dots can easily enter the cell and effectively remove ROS. Therefore, the carbon dots are thought to protect cells from oxidative damage by high levels of ROS.

## 1. Introduction

Carbon dots (CDs), as a kind of quasi-zero-dimensional carbon nanomaterial, have been widely studied by scientific researchers thanks to their unique advantages, and have broad application prospects in the fields of medical diagnosis, bioprobes, bioimaging, and tumor treatment [1,2,3,4]. CDs have also been reported to be used in the detection and elimination of reactive oxygen species (ROS), as a result of the special effect of ROS on organisms [5,6,7,8,9,10]. As we know, ROS is a class of single electron reduction product of oxygen in vivo [11]. The most common ROS species include singlet oxygen (O_2_), hydrogen peroxide (H_2_O_2_), hydroxyl radical (-OH), superoxide (O_2_^−^), nitric oxide (NO), and peroxynitrite (ONOO^−^), among others. Once ROS are produced, because of their high reactivity, they may cause significant damage to proteins/free amino acids, DNA/RNA, and lipids in cells [12,13,14,15]. The oxidation of protein by ROS can cause hydroxylation, nitrification, and chlorination of some groups or some amino acid side chains, leading to protein inactivation. ROS can also produce DNA damage, including coding errors, point mutations, single strand cleavage, abnormal amplification, and oncogene activation [16]. Interactions between ROS and DNA may lead to the formation of DNA–protein crosslinks, which accumulate in heart and brain tissue and may play a role in aging, cancer, and neurodegenerative diseases. Besides, excessive ROS levels will lead to lipid peroxidation of biofilm, resulting in loss of membrane fluidity, abnormal membrane potential, membrane rupture, and ultimately cell contents’ leakage [17]. Moreover, ROS may also promote the differentiation of tumor stem cells, promote epithelial mesenchymal transformation [18], and induce metabolic reprogramming to increase drug resistance in cancer cells [19]. In particular, many serious human diseases, such as cancer, diabetes, Alzheimer’s disease, Parkinson’s disease, cardiovascular disease, arthritis, and so on, are related to the damage of harmful high levels of ROS [20,21,22,23]. Thus, scavenging excess free radical is good for health and longevity of mankind.

Recently, Xu and others reported a kind of selenium-doped carbon quantum dots (Se-CQDs) for free-radical scavenging, which were synthesized from selenocysteine through hydrothermal treatment [24]. The Se-CQDs can effectively scavenge free radicals, thereby effectively protecting cells from oxidative stress. However, selenocysteine is a rare substance, and is expensive and not easily available. So, this method is not suitable for mass production of reducing carbon dots.

Herein, we replaced selenocysteine with lutein, a natural plant extract, to make carbon dots with strong reducing properties. As we know, lutein is widely present in plants such as vegetables, fruits, and flowers. It has a high reducing ability and can resist the damage of ROS [25]. However, it is almost insoluble in water, making it difficult to apply to cell protection. Fortunately, when lutein is made into carbon dots (L-CDs), the L-CDs have good water solubility and biocompatibility, and maintain the original reducing properties. In particular, through endocytosis, L-CDs become beneficial to enter the cells, thereby achieving the role of intracellular elimination of ROS (Scheme 1). Thus, L-CDs are considered suitable for mass production and have great application prospects in cell oxidation protection.

## 2. Materials and Methods

### 2.1. Materials and Instruments

All reagents and solvents were of analytical grade and have high purity available, but without deeper purification. Lutein was purchased from Energy Chemical (Shanghai, China), and Ethylenediamine was obtained from Aladdin Reagent (Shanghai, China). 1,1-diphenyl-2-picrylhydrazyl (DPPH) was purchased from Solarbio (Beijing, China). Ethanol was purchased from Shanghai Titan Scientific Co., Ltd. RPMI-1640 medium was obtained from GIBCO. WST-1 Cell Proliferation and Cytotoxicity Assay Kit, Reactive Oxygen Species Assay Kit containing 2’,7’-dichlorofluorescein diacetate (DCFH-DA) and ROSUP (ROS reagent, as a positive control for ROS level for cells) [26,27,28,29], penicillin, and streptomycin were all obtained from the Beyotime Institute of Biotechnology (Jiangsu, China). Fetal bovine serum (FBS) was obtained from Zhejiang Tianhang Biotechnology Co., Ltd.

Transmission electron microscopy (TEM) experiments were recorded by a JEOL-2100F transmission electron microscope (JEOL, Tokyo, Japan). Nano ZS/ZEN 3690 (Malvern, UK) was used to investigate the particle size distribution and surface potential of the CDs. The UV/vis absorption was measured by a UV/vis absorption spectrophotometer (Agilent Cary 300 Scan, Agilent Technologies, Inc., Santa Clara, California, USA). Fourier transform infrared (FT-IR) spectra were acquired using an FT-IR spectrometer (Agilent Cary 660, Agilent Technologies, Inc., Santa Clara, CA, USA). The fluorescence emission spectra of the carbon quantum dots were measured on a Hitachi F-4600 fluorescence spectrophotometer (Hitachi, Japan).

### 2.2. Preparation of L-CDs

Nitrogen-doped carbon dots from lutein were synthesized by the hydrothermal method, a facile and high-output strategy, under mild conditions. To begin with, 0.4 g of lutein was dissolved in the mixture of 20 mL of ethanol and 20 mL of ultrapure water. Once the lutein was completely dissolved through ultrasonic dissolving and constant stirring, 0.2 mL of ethylenediamine was added to the mixture. The mixture was then transferred into a 100 mL Teflon liner stainless steel autoclave and continuously heated at 140 °C for 12 h. After the autoclave was cooled to room temperature, the resultant solution was collected by removing the large particles by filtering through a 0.22 µm membrane. After first-time freeze-dry of the solution, the unpurified powder was obtained. The powder was dissolved again in ultrapure water, and centrifuged for 10 min at 10,000 rpm to remove water-insoluble lutein. The solution was then dialyzed (cut-off Mn: 1.0 kDa) in deionized water for 12 h to remove small molecules. Ultimately, the purified L-CDs powers were obtained through the second-time freeze-dry.

### 2.3. Quantum Yield (QY) Measurement

The QY was measured by the standard sample reference method [30,31,32]. Using quinine sulfate (QY  =  0.54) as the standard sample, the calculation formula is as follows:$$Ø\;=\;{Ø}_{r}\;\times \;\frac{I}{A}\;\times \;\frac{A_{r}}{I_{r}}\;\times \;\frac{n^{2}}{n_{r}^{2}}$$
where *Ø* is the QY, *I* is the measured integrated emission intensity, *n* is the refractive index, and *A* is the optical density. The subscript “*r*” refers to the reference fluorophore (quinine sulfate).

### 2.4. Cytotoxicity Assays

NCI-H1299 cells were selected for cytotoxicity testing of as-prepared L-CDs by WST-1 assay. NCI-H1299 cells were cultured in 96-well plates (8000 cells/well) under humidified 5% CO_2_ atmosphere at 37 °C for 24 h. The cell culture medium was RPMI-1640, containing 10% fetal bovine serum (FBS), 50 μg/mL streptomycin, and 50 unit/mL penicillin. There is 100 μL of medium per well. The cell culture medium was subsequently replaced with L-CDs solution of various concentrations, followed by incubation for 24 h. Then, 10 μL of WST-1 reagent solution was added into each well, and further incubated at 37 °C for another 40 min. The 96-well plate was then placed on a shaking table and shaken for one minute to fully mix the mixture. The absorbance at 450 nm of the mixture was recorded by the microplate reader, and the cell viability values were calculated according to the following formula: cell viability (%) = (the absorbance of experimental group/the absorbance of control group) × 100%.

### 2.5. Radical Scavenging Activity (RSA)

The scavenging activity of L-CDs to DPPH free radicals was assessed by monitoring the reduction of absorption at 520 nm of DPPH in ethanol solution. Each time, 10 μL of L-CDs solution was added to 2 mL of DPPH ethanol solution (100 μM). The decrease of the absorption peak at 520 nm was recorded.

Moreover, the capability of L-CDs to scavenge reactive oxygen species in cells with reactive oxygen species assay was also investigated. Similar to the operation of cytotoxicity assays, as mentioned above, NCI-H1299 cells were cultured in 96-well plates (8000 cells/well) under humidified 5% CO_2_ atmosphere at 37 °C for 24 h. The cell culture medium was subsequently replaced with L-CDs solution of various concentrations, followed by incubation for 12 h. Then, 1 μL of ROSUP (5 mg/mL) was added into some wells as the positive control, followed by incubation for 1 h. After that, the cell culture medium was replaced with DCFH-DA (10 μM/L), followed by incubation for 20 min. Subsequently, the solution was removed, and the plates were washed three times with medium without fetal bovine serum. Finally, the fluorescence intensity at 525 nm was recorded with a microplate reader, upon the excitation of 488 nm.

## 3. Results and Discussion

### 3.1. Synthesis and Characterization of L-CDs

The water-soluble lutein-based carbon dots were synthesized as shown in Figure 1A by the hydrothermal method. The prepared L-CDs present a uniform spherical shape and favorable dispersion, and the TEM image shows the size of the carbon dots is about 40 nm (Figure 1B). The dynamic light scattering (DLS) result confirmed the diameter of carbon dots (inset of Figure 1B). The zeta potential of L-CDs was measured as −13.3 mV, which is probably owing to the rich phenolic hydroxyl groups on the surface of L-CDs. In addition, FT-IR spectra were applied to characterize chemical composition and surface functional groups of L-CDs. As shown in Figure 1C, the absorption bands at about 1320 cm^−1^ and 1560 cm^−1^ represent the stretching vibrations of C=C-H, and the peak at 2970 cm^−1^ is attributed to the N-H vibrations. In the meantime, the shape peak at 1050 cm^−1^ and 1646 cm^−1^ correspond to the C-N and C=O vibrations, respectively. The above results indicate that the surface of the L-CDs contains both amino groups and hydroxyl groups; at the same time, the L-CDs are rich in unsaturated double bonds, which is also the reason the L-CDs have super reducibility.

### 3.2. The Photoluminescence Properties of L-CDs

The optical properties of L-CDs were recorded by UV/vis absorption spectrophotometer and Hitachi F-4600 fluorescence spectrophotometer, respectively. The UV/vis absorption spectrum shows that the L-CDs have a broad absorption band ranging from 200 nm to 500 nm (Figure 2A), and at the same time, there is a shoulder at 325 nm, which may be a π–π * transition from a similar aromatic structure. Upon the irradiation of 365 nm, the L-CDs exhibit bright blue fluorescence. The fluorescence spectrometer shows that the fluorescence emission peak of L-CDs locates at 460 nm, whose corresponding excitation peak is at 370 nm (Figure 2B). In addition, like most other CDs, the as-prepared L-CDs also exhibit excitation dependent fluorescence properties. Under the excitation light of 370 nm, L-CDs have the strongest fluorescence emission. With the increase of excitation wavelength (from 370 to 460 nm), the fluorescence wavelength shifts red and the fluorescence intensity decreases gradually (Figure 2C). Using quinine sulfate as reference material, the QY of L-CDs is calculated to be about 10%. Using quinine sulfate as reference material, the QY of L-CDs is calculated to be about 10%. Compared with the reported carbon dots, the QY of the L-CDs is not high, which may be because of the fact that the carbon dots have a wide absorption spectrum without an obvious absorption peak, which leads to the low excitation efficiency. Another reason may be that the carbon dots are prepared by strong reducing materials. The surface of carbon dots is rich in unsaturated flexible carbon chain structures, and the excited state energy of L-CDs is easily consumed by the vibration and rotation of the surface branched chains, thus reducing the fluorescence quantum efficiency.

Moreover, the stability of the prepared L-CDs was measured. Figure 3A gives the effect of pH on the fluorescence intensity of L-CDs. At pH 7 and 8, the L-CDs have the strongest fluorescence intensity; under alkaline conditions, the fluorescence intensity decreases slightly (at pH 12, the relative fluorescence intensity only decreases by less than 10%); While under acidic conditions, the fluorescence intensity decreased significantly, especially at pH 1, and the relative fluorescence intensity drops to 60%. In general, the L-CDs have good fluorescence stability under neutral and alkaline conditions, which is probably related to the rich amino groups on the surface of CDs. At the same time, the stability of L-CDs with time was also measured. For 2 weeks, the fluorescence intensity of L-CDs was recorded, and the solution in the centrifuge tube was photographed under UV light. As shown in Figure 3B, at the beginning of preparation, L-CDs had a bright blue to green fluorescence, and after two days, the fluorescence intensity almost did not change; after 14 days of storage, the fluorescence decreased slightly (about 15%) and the fluorescence became bluer. This conclusion is consistent with other articles. A large number of studies have shown that blue fluorescent carbon dots have good stability and can be stored for more than one month [4,33,34,35]. It is speculated that the blue emission is the intrinsic fluorescence of L-CDs, which may come from the graphene-like structure of L-CDs themselves and will not be changed by the surface functional groups.

### 3.3. Cytotoxicity Test of L-CDs

L-CDs have potential as biomedical agents on account of their excellent biocompatibility and strong photoluminescence. Therefore, it is necessary to evaluate their cytotoxicity. The standard WST-1 assay utilizing NCI-H1299 cell was performed as the model to achieve the above objectives. As depicted in Figure 4, the L-CDs exhibit extremely low cytotoxicity. After the addition of L-CDs (1.5 mg/mL) for 24 h, NCI-H1299 cells still present viability even higher than 80%. Under the lower level of L-CDs (less than 100 μg/mL), the viability of cells is about 100%. These data clearly demonstrate the good biocompatibility of L-CDs.

### 3.4. Free Radical Scavenging in Solution

DPPH was used to investigate the radical scavenging activity of L-CDs in solution. DPPH is a very stable free radical with a nitrogen center. After receiving a hydrogen radical, DPPH could form the steady DPPH_2_ complex, bringing the decrease of the absorption peak at 520 nm [36]. The scavenging activity of L-CDs towards DPPH at the concentration from 0 to 160 μg/mL was investigated. As demonstrated in Figure 5, as the concentration of L-CDs continues to increase, the absorbance of DPPH at 520 nm gradually decreases. Ultimately, the ratio of the radical scavenging activity of L-CDs towards DPPH was calculated to be about 31.5%. The calculation method follows this formula: Inhibition (%) = (the absorbance of A_control_—the absorbance of A_sample_/the absorbance of A_control_) × 100%, where A_sample_ and A_control_ refer to the absorbance of the DPPH at 520 nm with and without L-CDs, respectively.

### 3.5. Application of Free Radical Scavenging in Cells

Reactive Oxygen Species Assay Kit was used to investigate the intracellular radical scavenging activity of L-CDs. DCFH-DA is a cell-permeable probe for detecting the intracellular ROS. DCFH-DA has no fluorescence itself with an ability to cross the cell membrane freely. After entering the cell, DCFH-DA is hydrolyzed to DCFH by the cellulase, and then intracellular ROS can oxidize non-fluorescent DCFH to fluorescent DCF. Then, the intracellular ROS level can be determined by measuring the fluorescence intensity of DCF. In this experiment, we detected the scavenging effect of L-CDs on ROS of cells themselves, and examined the ability of L-CDs to remove the intracellular elevated ROS produced by ROSUP treatment (ROSUP is a positive contrast agent that can produce extra ROS in cells). As shown in Figure 6, without ROSUP, the fluorescence intensity of cells decreased slightly after adding L-CDs, which indicated that the ROS level in normal cells was not high and could be effectively removed by L-CDs. Meanwhile, with ROSUP, the level of intracellular ROS increased significantly, as a result of the introduction of external ROS. When the L-CDs were added, the fluorescence intensity of the cells decreased sharply; even 0.05 mg/mL of L-CDs can reduce the fluorescence intensity by about 30%, and 0.25 mg/mL of L-CDs can further reduce the fluorescence intensity, making it almost no different from the normal cell. The results prove that a low concentration of L-CDs solution can significantly remove extra ROS in cells. Thus, the L-CDs have great application prospects in cell antioxidant.

## 4. Conclusions

In summary, we have fabricated fluorescent nitrogen-doped carbon dots from lutein by the hydrothermal method, which is a facile, eco-friendly, and high-output strategy. The as-prepared L-CDs exhibit bright blue fluorescence, which enables intracellular imaging. At the same time, the L-CDs have good water solubility and biocompatibility, and do not have any toxicity to cells. Through endocytosis, L-CDs can easily enter cells, thereby eliminating free ROS in cells and protecting cells from oxidative damage. L-CDs are considered suitable for mass production and have great application prospects in cell oxidation protection.

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
