# Peer review of "Green Synthesis of Lutein-Based Carbon Dots Applied for Free-Radical Scavenging within Cells"

_materials, 2020, doi:10.3390/ma13184146_

Round 1

Reviewer 1 Report

In the manuscript entitled "Green Synthesis of Lutein-Based Carbon Dots
Applied for Free-Radical Scavenging within Cells" the authors have synthesized the nitrogen-doped carbon dots from lutein by hydrothermal method. They used the carbon dots mainly in the search of free-radicals within the cells. The experimental design looks good to me. But there are some flaws in the manuscripts as outlined below:

1. The introduction section is not flowing continuously. It needs to rewritten. It is better to write the story beginning with the carbon dots introduction, followed by the importance of free radical scavenging and concluding with the significance of their research.  

2. The authors have found that after 14 days of storage, the fluorescence intensity of the L-CDs decreases by less than 15%.  No satisfactory reasons for this cause have been given so far. 

3. Carbon dots are usually excitation dependent (with some exceptions). In this manuscript, the authors have only used one excitation. I suggest the authors to check whether their carbon dots is excitation dependent and provide a separate figure instead of figure 2B. 

4. It would have been better if the authors provided the quantum yield for their carbon dots. This would significantly enhance the quality of manuscript. 

Reviewer 2 Report

Comments to the author:

In this manuscript, the authors presented the novel fabrication of carbon dots based on a green process using lutein (L-CDs) by an hydrothermal method. Besides, they presented this material as an alternative to selenium-doped carbon quantum dots which can protect cells from oxidative stress but its industrial production is not affordable.The authors demonstrated the biocompatibility of the L-CDs and they effectively demonstrated that by endocytosis, L-CDs can eliminate ROS in cells and hence they possess a protective role from oxidative damage. Overall, I would recommend the publication of this contribution but some issues should be addressed.

The following are some questions and suggestions for improving their work:

Major issues:

  1. ROSUP compound was not defined in the whole manuscript. What is it exactly?
  2. The authors showed the cell viability after incubation with L-CDs for 24h. As this particles are though to protect cells against ROS it would be interesting to know what is the long-term the cell viability.
  3. According to the literature, is a 30% of reduction of ROS sufficient or comparable to other works?

Minor issues:

  1. Absorbance has no units. All the figures where absorbance is in arbitrary units should be corrected.
  2. What is the pH inside the cells? Could be lower than 5?

Reviewer 3 Report

This paper describes the preparation of carbon dots (CD) from lutein and ethylenediamine and its use in the removal of intercellular ROS. The preparation of the CD follows a standard synthetic strategy with the appropriate purification procedure.

  • However, the reactivity of CD towards ROS has already been described (for example in: DOI: 10.1016/j.snb.2015.06.072, 10.1016/j.aca.2017.01.007). This subject should be briefly review in the introduction.
  • The average size of the CD, about 40 nm, is too big taking into consideration that the normal size of CD is of a few nanometers. This must be further studied.
  • Moreover, the results described in section 3.5, which are the most important of the paper, accordingly to its objective are not satisfactorily because: (1) the number of results are very few; (ii) from the analysis of fig. 6 we conclude that statistically the results are not different – the standard deviations overlapped.
  • Further results are needed and a statistically analysis is required.

Round 2

Reviewer 1 Report

The revised version looks fine to me.

Reviewer 3 Report

The authors have answered satisfactorily to the referee suggestions.